# Boron Delivery to Brain Cells via Cerebrospinal Fluid (CSF) Circulation in BNCT of Brain-Tumor-Model Rats—Ex Vivo Imaging of BPA Using MALDI Mass Spectrometry Imaging

**DOI:** 10.3390/life12111786

**Published:** 2022-11-04

**Authors:** Sachie Kusaka, Yumi Miyake, Yugo Tokumaru, Yuri Morizane, Shingo Tamaki, Yoko Akiyama, Fuminobu Sato, Isao Murata

**Affiliations:** 1Division of Sustainable Energy and Environmental Engineering, Graduate School of Engineering, Osaka University, Yamadaoka 2-1, Suita 565-0871, Osaka, Japan; 2Forefront Research Center, Graduate School of Science, Osaka University, 1-1 Machikaneyama, Toyonaka 560-0043, Osaka, Japan

**Keywords:** boron neutron capture therapy (BNCT), cerebrospinal fluid (CSF) circulation, brain tumor, melanoma model rat, ex vivo imaging, matrix-assisted laser desorption/ionization (MALDI) mass spectrometry imaging (MSI)

## Abstract

The blood–brain barrier (BBB) is likely to be intact during the early stages of brain metastatic melanoma development, and thereby inhibits sufficient drug delivery into the metastatic lesions. Our laboratory has been developing a system for boron drug delivery to brain cells via cerebrospinal fluid (CSF) as a viable pathway to circumvent the BBB in boron neutron capture therapy (BNCT). BNCT is a cell-selective cancer treatment based on the use of boron-containing drugs and neutron irradiation. Selective tumor targeting by boron with minimal normal tissue toxicity is required for effective BNCT. Boronophenylalanine (BPA) is widely used as a boron drug for BNCT. In our previous study, we demonstrated that application of the CSF administration method results in high BPA accumulation in the brain tumor even with a low dose of BPA. In this study, we evaluate BPA biodistribution in the brain following application of the CSF method in brain-tumor-model rats (melanoma) utilizing matrix-assisted laser desorption/ionization (MALDI) mass spectrometry imaging (MSI). We observed increased BPA penetration to the tumor tissue, where the color contrast on mass images indicates the border of BPA accumulation between tumor and normal cells. Our approach could be useful as drug delivery to different types of brain tumor, including brain metastases of melanoma.

## 1. Introduction

Boron neutron capture therapy (BNCT) is a cell-selective cancer treatment based on the high likelihood of boron-10 to cause nuclear reactions with neutrons [1,2]. BNCT is the only radiation therapy characterized by this feature. Boronophenylalanine (BPA) is widely used as a boron drug for BNCT in malignant melanoma and brain tumors as well as in recurrent head and neck cancer [3]. BPA is imported into the cells by the L-type amino acid transporter (LAT1), which is lowly expressed at the blood–brain barrier (BBB) [4]. However, developing an efficient and effective BPA-to-brain tumor delivery system for BNCT is a current challenge. Selective biodistribution and accumulation of large amounts of BPA into the brain tumor, rather than normal tissue, are required for successful BNCT [5].

In general, three different strategies for delivery of drugs to the brain have been attempted: (1) extensive opening of the BBB (paracellular approach), (2) delivery across the BBB and (3) circumventing the BBB (transcellular approach) [6,7]. In approach (1), focused ultrasound (FUS) with microbubbles has been proved to transiently open the BBB, allowing the penetration of administered drugs into the brain [8]. However, the opening of the BBB ultrasound remains only for a short period of time and therefore the procedure must be repeated in multi-fractionated drug delivery schemes. In approach (2), drugs are encapsulated in carriers such as liposomes or proteins [9,10,11,12]. These drug carrier devices promote drug transport across the BBB in a mediated transcytosis pathway or the BBB endogenous-carrier system. However, due to the finite number of receptors in vivo and the limited payload of drug carriers, the drug carrier approach has poor drug transport [13]. In approach (3), the drug is directly delivered to the brain via either an intracerebroventricular injection, continuously injecting the drug into the interstitial spaces of the brain or an intracerebral implantation. This approach can achieve therapeutic concentrations in brain tumors by continuous infusion. Although promising as an effective drug delivery method in concept, the administration of drugs via cerebrospinal fluid (CSF) or brain interstitial fluid has been considered challenging due to limited drug distribution to the brain. [8,14]. However, the idea of drug delivery to brain cells via CSF microcirculation is being reconsidered as an important development in the neuroscience field (the detail will be described later). Our laboratory has been developing a system for boron delivery that meets the requirements of BNCT, and we call the system that delivers boron drugs to brain cells via the CSF “boron cerebrospinal fluid administration method” [15,16]. In this study, we aim to visually evaluate BPA biodistribution, especially the tumor selectivity of BPA, following the CSF administration method in brain-tumor-model rats (melanoma).

CSF is a specialized fluid produced by the choroid plexus in the lateral ventricle. Drug concentrations in CSF are commonly used to estimate drug penetration into the brain. However, the interaction between CSF and brain cells is complex and CSF is not necessarily representative of the interstitial environment of the brain [17,18]. Currently, the potential of harnessing CSF circulation for drug delivery to deep areas of the brain remains a topic of discussion. For decades, CSF circulation was thought to be only useful for reaching tissues in the immediate vicinity of the CSF circulatory system [19]. Now, it is recognized that the CSF also flows through the parenchyma via a mechanism referred to as the CSF microcirculation [20]. In recent years, researchers have focused on CSF circulation, instead of blood circulation, for drug delivery to the brain [21,22,23]. This approach could be useful as a strategy that circumvents the BBB when delivering drugs to different types of brain tumors, including brain metastases of melanoma [24]. In our previous study, we found that boron tends to accumulate more quickly and be excrete more smoothly from normal cells in Slc:SD rats when injected according to the CSF administration method compared with the intravenous (IV) administration method [15]. Consequently, the CSF administration method resulted in a high T/N ratio, which is the ratio of boron concentration in tumor cells (T) to normal cells (N), comparable to that of the IV administration method. Additionally, the CSF administration method resulted in high boron accumulation in tumor cells even for a low BPA dose (1/10 of the dose used in the IV administration method). The CSF-based method may thus enable low-dose treatment, leading to lower toxicity and costs than the IV administration method.

We conducted ex vivo imaging of BPA in thin brain-tissue slices of brain-tumor-model rats using matrix-assisted laser desorption/ionization mass spectrometry imaging (MALDI-MSI) to validate our findings. First, we performed a time-course study of BPA biodistribution in the brain parenchyma following administration of BPA to the lateral ventricle of brain-tumor-model rats. Second, we compared the characteristics of BPA biodistribution in brain tumors for IV administration vs. administration via CSF circulation.

## 2. Materials and Methods

### 2.1. Study Design

The animal study protocol was approved by the Animal Care and Use Committee for Osaka University (protocol code 2019-1-3).

We prepared the brain-tumor-model rats (melanoma) and administered the boron compound via the CSF or IV administration method. Brain samples were excised into thin slices using a cryostat microtome and subjected to mass spectrometry analysis to obtain boron macro images (Figure 1)

### 2.2. B16F10 Melanoma Model Rats

The animals were eight-week-old male Slc:SD rats weighing approximately 250~350 g. The B16F10 tumor cells were provided by Tohoku University, and the implantation into the rat brains was conducted by CLEA Japan, Inc. (Tokyo, Japan). The B16F10 cells adjusted to 1.0 × 104 cells/3 µL were injected for 3 min at a rate of 1 µL/min at 3 mm right lateral, 0 mm caudal, and 4.5 mm ventral to the rat brain bregma. The rats with brain tumors were supplied for the present experiment 12 days after implantation.

### 2.3. Boron Compound

4-Borono-L-phenylalanine (Sigma–Aldrich Corp., Saint Louis, MO, USA) was prepared as a fructose complex to increase its water solubility (20 mg/mL BPA). The boron concentration in this solution was determined as 1219 ppm using inductively coupled plasma atomic emission spectroscopy (ICP–AES) based on a boron standard solution [25].

### 2.4. Boron CSF Administration to Brain Tumor Rats

Rats were anesthetized by inhalation of 3% isoflurane at an adjustable flow rate of 1.5 L/min and an intramuscular injection of ravonal (0.15 mg/kg). We prepared three rats for the stereotaxic brain operation using a stereotaxic instrument (SR-5R-HT, Narishige International Ltd., Tokyo, Japan), and introduced a guide cannula (C313G, Bio Research Center Co. Ltd., Aichi, Japan) into the medium aperture of lateral ventricle (0.8 mm caudal and 7 mm ventral to the rat brain bregma). BPA was administered through the guide cannula at 4 mg/kg/h for 30 or 160 min to rats with brain tumors. Brain samples were collected from each rat at 0 or 60 min after the end of the infusion (Figure 2, Rat A–C). As a control specimen, brain tissue was collected from a non-tumor rat to which BPA was administered. The obtained brain samples were preserved in isopentane solution in a freezer until mass spectrometry imaging analysis. The CSF samples were collected by cisternal puncture 30 or 60 min after the start of the infusion to verify BPA delivery to the CSF. CSF samples were stored in a freezer until the measurement of boron concentrations with ICP–AES (ICPS-8100, Shimadzu Corp., Kyoto, Japan).

### 2.5. Intravenous (IV) Administration to Brain Tumor Rats

The anesthesia and brain operations were performed in the same way as in the experiments for the CSF administration method for comparison. BPA as administered to one rat via the cervical vein at 350 mg/kg/h for 120 min. The administration protocol was based on the veterinary protocol for canines in BNCT with adjustments [26]. The rat brain tissue was immediately collected 60 min after the infusion ended and preserved in isopentane solution until MALDI-MSI measurement (Figure 2, Rat D).

### 2.6. Preparation of Samples for MALDI-MSI

Brain samples obtained as described in Section 2.4 and Section 2.5 were excised into 16 µm thick slices using a cryostat microtome (CM1100, Leica Biosystems, Nusslosh Germany), and each slice was placed on a stainless plate (50 μm thickness, 20 mm × 20 mm, Iwata industry, Tokyo, Japan). The solution of matrix reagent, 2, 5-dihydroxybenzoic acid (40 mg/mL) was sprayed onto the brain slice using an airbrush sprayer.

### 2.7. MALDI-MSI

The mass spectra of brain specimens were recorded using a MALDI-SpiralTOF/TOF mass spectrometer (JMS-S3000, JEOL Ltd., Akishima, Japan). The MSI scanning for the brain section was carried out at 60 μm spatial resolution (60 μm pixel size) and analyzed using msMicroImager Extract software (ver. 3.0.0.1, JEOL Ltd., Akishima Japan). The mass image was obtained from the peak of protonated BPA molecule based on the peak intensity in each pixel of the sample to generate estimations based on the BPA macro images. The details of conditions used in mass spectrometry imaging are as described in our previous report [27].

### 2.8. Ensuring Image Reliability

Our research group has succeeded in drawing an image with [BPA + H]^+^, which is a protonated BPA molecule (Figure 3). The peak of [BPA + H]^+^ at *m/z* 210.084 was detected separate from adjacent peaks with high mass resolution (Figure 3d). The difference between the observed *m/z* value of [BPA + H]^+^ and the calculated value (*m/z* 210.093) was within acceptable limits (<50 ppm error). The spotted BPA standard on the melanoma or cortical region was observed at *m/z* 210.084 and 210.085, respectively. The peaks at *m/z* 210.026, 210.063 and 210.098 clearly imaged the tumor morphology in every melanoma sample (Molecular species not identified). Therefore, the possible overlap with [BPA + H]^+^ and tumor-derived ions at *m/z* 210.084 was evaluated by comparing the distribution in the mass image obtained from the peak at *m/z* 210.084 with that of peaks at *m/z* 210.026, 210.063 and 210.098. As can be seen, Figure 3d showed the distribution of BPA within tumors, which is different from other peaks derived from tumors at m/z 210.026, 210.063 and 210.098. (Figure 3b,c,e). Moreover, if BPA ion species other than [BPA + H]^+^ (for example [BPA − H_2_O + Na], (Figure 3f) are detected, we ensured the similarity of distribution in both images of [BPA + H]^+^ and [BPA − H_2_O + Na]^+^. The mass image of each peak at *m/z* 209.914 or *m/z* 307.042 showed uniform distribution within the whole area of the brain section, indicating that it was considered to be endogenous biomolecules (Figure 3a) (Molecular species not identified).

## 3. Results

Figure 4 show the right-side brain images of the non-tumor rat (control) after BPA administration via the CSF method for 160 min. 

Figure 5 show the right-side brain images of the brain-tumor-model rat (Rat A) after BPA administration via the CSF method for 30 min.

Figure 6 show the right-side brain image of the brain-tumor-model rat (Rat B) after BPA administration via the CSF method for 160 min.

Figure 7 show the right-side brain images of the brain-tumor-model rat (Rat C) at 60 min after BPA administration via the CSF method for 160 min.

Figure 8 show the right-side brain images of the brain-tumor-model rat (Rat D) at 60 min after BPA administration via the IV method for 120 min, cited from Miyake’s paper [27]. 

Our study results demonstrate that a considerable amount of BPA is localized in tumor cells after administration via CSF circulation (Figure 6). The difference in color contrast between tumor and normal cells was much more pronounced in the image corresponding to the longer administration time of 160 min (Figure 6) than that of 30 min (Figure 5). Moreover, at 60 min after the end of the infusion, we only observed BPA along the lateral ventricle and not in the tumor (Figure 7). In our comparison of the CSF and IV methods, both demonstrated that BPA is localized in the invasive margin of the brain tumor and demarcates the border between tumor and normal cells. Moreover, our mass images show the non-uniformity of BPA accumulation in the tumor, even within a small tumor 5 mm in diameter (Figure 6 and Figure 8).

## 4. Discussion

BPA, which is one of the boron-containing delivery drugs available for BNCT, is actively uptaken into cells through LAT1, which is responsible for transporting large neutral amino acids [28]. LAT1 overexpression is observed in a variety of tumor cells; therefore, BPA is well suited for transport of boron into tumor cells in BNCT [29]. BPA is now commonly administered via the blood vessels for BNCT because some LAT1 expression occurs at the BBB [30]. However, compared with other anticancer drugs, a much higher degree of accumulation of boron-containing drugs into tumor cells is required to ensure the efficacy of BNCT, whereby nontumor cells are left mostly unharmed [31,32]. Our laboratory has been developing an alternative system for efficient boron delivery to brain cells via CSF. The CSF-based method may enable low-dose treatment, which might be advantageous in terms of patient safety during BNCT compared with the IV method.

In our study, we examined the BPA biodistribution in rat brain after BPA infusion via CSF circulation to demonstrate the usefulness of the CSF administration method for brain tumors. As a result, we generated mass images of MALDI-MSI that demonstrate that the CSF approach enables deeper BPA delivery into the brain parenchyma; therefore, our results may potentially contribute to the further development of neuroscience studies.

### 4.1. Previous Research and Progress of CSF Administration Method

Our previous studies have confirmed sufficient boron concentrations in the brain tumor following CSF administration, despite low doses of BPA [15]. In addition, the BPA biodistribution image obtained in the present study showed that BPA accumulates evenly in the deep part of the brain tumor, rather than a biased BPA biodistribution due to diffusion from CSF. These results suggested that CSF administration method can contribute to the development of BNCT for patients with brain tumor.

Through the studies thus far, apart from aiding in the development of BNCT, important contribution to the neuroscience field was considered to be possible. In our previous studies, we investigated BPA kinetics via CSF in normal rats [16]. It was found that a continuous infusion of BPA into the CSF for at least 60 min is required to saturate the boron concentration in the CSF, and that boron uptake into brain cells during BPA-saturated CSF has little correlation with the dose of BPA. Furthermore, after the end of administration, in normal rats, the brain boron concentration time-to-peak was found to be not immediate, but rather at 60 min after the end of the infusion to the lateral ventricle, whereas the boron concentration in the CSF decreased promptly after the end of infusion. In the present experiment, we visually confirmed BPA accumulation in the brain parenchyma with a 160-min administration via CSF and BPA excretion from the brain parenchyma after 60 min. These results suggest that the uptake of BPA into the brain parenchyma is not related to the total amount of BPA, but rather to the time after the start or end of infusion. Moreover, this may be related to the time that the CSF compartment maintains positive pressure during and after drug administration.

### 4.2. Time-Course Study of BPA Biodistribution following CSF Administration

(Distribution) We performed a time-course study of BPA biodistribution following administration via CSF in the brain parenchyma. We observed an increased BPA penetration of the tumor tissue after administration for 160 min (Figure 6) compared with 30 min (Figure 5). Our results are generally consistent with those of the literature (Naseri Kouzehgarani G et al.) showing that antibody penetration of the brain parenchyma via the periarteriolar spaces starting within 30 min of ABT-806 administration, with additional penetration after 45 and 60 min of administration [20]. We estimated that over 60 min of drug administration is required for penetration throughout the entire brain tissue after infusion (Details were explained in Section 4.1). 

(Excretion) At 60 min after the end of infusion, we observed BPA only along the lateral ventricle, while no BPA was observed in the tumor (Figure 7). However, we observed a small amount of BPA in normal tissue. Our results indicate that almost all BPA tended to be excreted from the brain cells to CSF within 60 min post-injection. In our previous study [15], we observed that the boron concentration peak in normal cells following CSF administration occurred at 60 min post-BPA injection. Even after 120 min, we detected small amounts of boron in normal cells. The present results could suggest that BPA would be accumulated quickly and effectively in the tumor cells, however, the BPA would be excreted more quickly from the tumor cells than normal cells.

### 4.3. Confirmation of the Potential of CSF Microcirculation

“Microcirculation” refers to CSF circulation through spinal or brain tissues and has often been referred to as “glymphatic” circulation in recent publications [33]. Previously, drugs injected into the CSF were thought to be distributed in blood and the brain surface but not in the brain parenchyma at distances exceeding 1–2 mm from the CSF compartment [30]. Recently, advances in ex and in vivo imaging and the advent of microdialysis have led to a strengthening consensus that a CSF “microcirculation” system, commonly referred to as the “glymphatic” system, exists within brain tissues [33,34,35,36]. CSF microcirculation through perivascular spaces ensures the uniform distribution of a drug throughout the entire brain and not only in tissues in the immediate vicinity of the CSF circulatory system [19,37].

The results of our previous study showed that the BPA spreads throughout the entire brain tissue within 60 min of administration via CSF [16]. In our present study, we observed vastly deeper penetration of BPA in the brain parenchyma, which was independent of the distance from the CSF circulatory system, after 30 or 160 min of administration via CSF. We ultimately conclude that the results imply that BPA injected via CSF arrives into the entire brain parenchyma via CSF microcirculation through perivascular spaces rather than through diffusion from CSF to a limited brain area.

### 4.4. BPA Biodistribution at the Cellular Level Based on MALDI-MSI

BNCT has the potential to selectively kill cancer cells at a cellular level. BNCT’s therapeutic efficacy is strongly dependent on the μm order boron biodistribution [38]. Selective tumor targeting with minimal normal tissue toxicity is required for boron-containing drugs in BNCT. Furthermore, brain cells play a critical role in the human body, and any damage to healthy cells can cause serious and permanent harm to the patient. In this study, BPA color gradients on the image, following CSF administration, were observed in the border between tumor and normal cells. Our results are comparable to those of the IV administration method, demonstrating the promise of this approach via CSF circulation for the treatment of patients with brain tumors. An evaluation of the characteristics of boron uptake into tumors is essential for patient dose management. Some methods facilitating the measurement of boron concentrations have been developed for basic research [39], in addition to methods for boron macro imaging and analysis, including neutron capture radiography (NCR) [40], secondary ion mass spectrometry (SIMS) [41], and electron energy-loss spectroscopy (EELS) [42]. Their spatial resolution can reach the μm range, which means they are capable of detecting boron at cellular and subcellular levels. However, these techniques are not able to generate images suitable for assessing the ratio of boron concentrations in tumor cells (T) to normal cells (N) (T/N ratio) for BNCT because the image area is too small to allow evaluation of this ratio. Recently, MALDI-MSI has become increasingly popular in both medical and basic research after the discovery of detailed histomolecular signatures in tissues, such as protein, lipid, metabolized, and drug, without the need for tagging or target-specific antibodies [43]. Our team developed an advanced technique for brain BPA macro imaging at a 60 μm spatial resolution with MALDI-MSI and successfully obtained BPA-uptake images in a brain-tumor-model rat [27]. This technique has helped us to evaluate the utility of the CSF administration of BPA, which is the first time such a result has been demonstrated.

The mass images with MALDI-MSI showed non-uniform BPA accumulation in the tumor following CSF administration, even within small tumors only 5 mm in diameter. The heterogeneity of BPA distribution in the tumor tissues suggests the possibility of tumor recurrence for patients because the area with a lower penetration of BPA might receive an insufficient dose for killing tumor cells in BNCT. To confirm the hypothesis, an improvement of quantitative performance of the MALDI-MSI technique is required such that is becomes possible to determine the absolute concentration of boron in normal and tumor cells, information that can be applied toward determining an appropriate dose for administration [44].

## 5. Conclusions

In this study, we examined the BPA biodistribution in the brain of a rat following an infusion of BPA via CSF circulation toward demonstrating the usefulness of the CSF administration method for brain tumors. We succeeded in capturing the characteristics of BPA behavior in the tumor when administering BPA to the lateral ventricles in the rat models. We obtained evidence that BPA arrives directly into the brain parenchyma with CSF microcirculation through perivascular spaces by creating images of BPA penetration into the brain parenchyma following administration via CSF. We believe that this approach, via CSF, is promising in ensuring delivery of BPA to deeper areas of the brain in BNCT. Using this method, administration should be applied for over 60 min to ensure penetration of the entire brain tissue. The above results are an example using brain tumor-model rat. Although the methods illustrated may not be applicable in all cases, they will potentially aid in developing a deeper understanding of drug delivery systems via cerebrospinal fluid. We must further develop the MALDI-MSI technique to measure the absolute concentration of BPA ex vivo imaging in BNCT studies of brain tumor patients, which will result in a lower tumor recurrence rate.

## Figures and Tables

**Figure 1 life-12-01786-f001:**
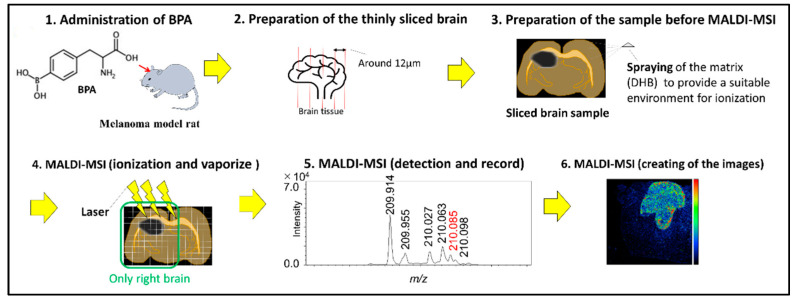
Schematic representation of the methods.

**Figure 2 life-12-01786-f002:**
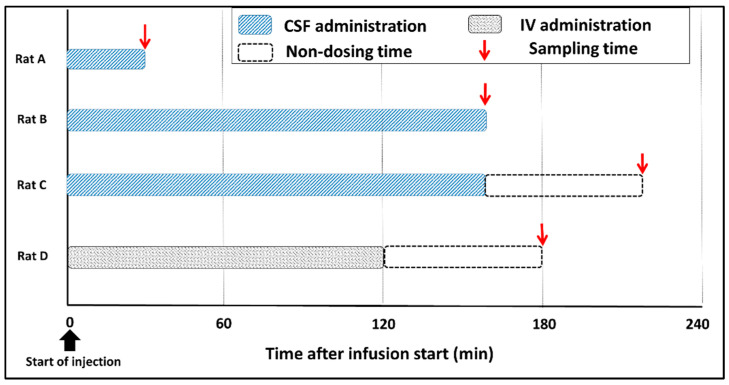
Summarizes of the administration setup.

**Figure 3 life-12-01786-f003:**
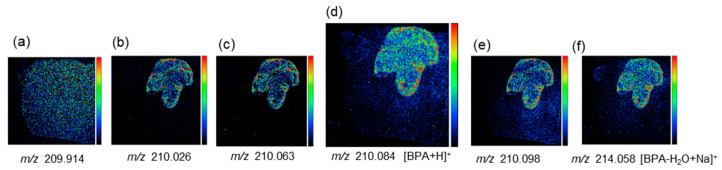
Mass images drawn with a protonated BPA molecule and its neighborhood spectra (Referenced from Miyake’s paper [27]). (**a**) The mass image of the peak at *m/z* 209.914. (b) The mass image of the peak at *m/z* 210.026. (**c**) The mass image of the peak at *m/z* 210.063. (**d**) The mass image of the peak at *m/z* 210.084. (**e**) The mass image of the peak at *m/z* 210.098. (**f**) The mass image of the peak at *m/z* 214.058.

**Figure 4 life-12-01786-f004:**
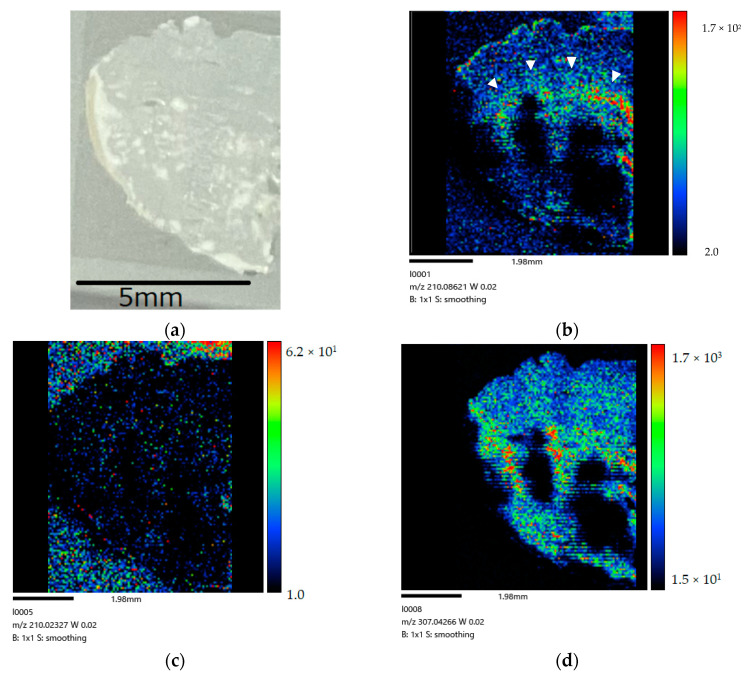
(**a**) Coronal section of Slc:SD rat after 160 min administration. (**b**) BPA biodistribution image generated by MALDI-MSI (drawn with [BPA + H]^+^ (*m/z* 210.084)). A large amount of BPA is present in the lateral ventricle (➣). The distribution in the image shows a different distribution from the tumor-derived peak (**c**) and the endogenous biomolecule-derived peak (**d**) following CSF administration. (**c**) The mass image obtained from peaks at *m/z* 210.026 originated in tumor. (**d**) The mass image obtained from peaks at *m/z* 307.042 originated from endogenous biomolecule.

**Figure 5 life-12-01786-f005:**
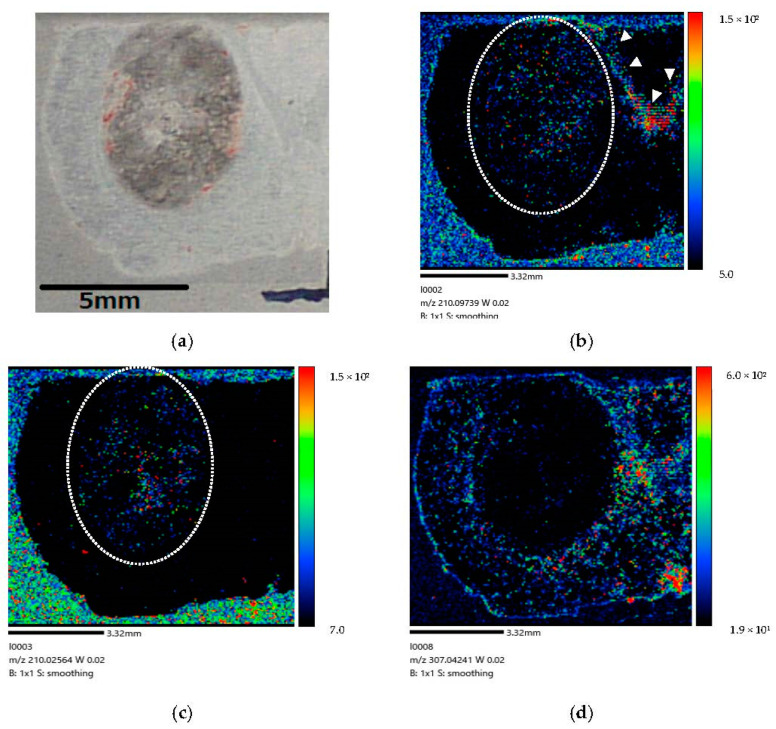
(**a**) Coronal section of brain-tumor-model rat after 30 min administration. (**b**) BPA biodistribution image generated by MALDI-MSI (drawn with [BPA + H]^+^ (*m/z* 210.097)). The dotted line (…..) indicates where the tumor is expected to exist. BPA penetration is observed in the brain tumor cells. A large amount of BPA is present in the lateral ventricle (➣). A small amount of BPA penetrates the brain tumor cells within 30 min following administration into the lateral ventricle of brain-tumor-model rat. (**c**) The mass image obtained from peaks at *m/z* 210.026 originated in tumor. The dotted line (…..) indicates where the tumor is expected to exist. (**d**) The mass image obtained from peaks at *m/z* 307.042 originated from endogenous biomolecule.

**Figure 6 life-12-01786-f006:**
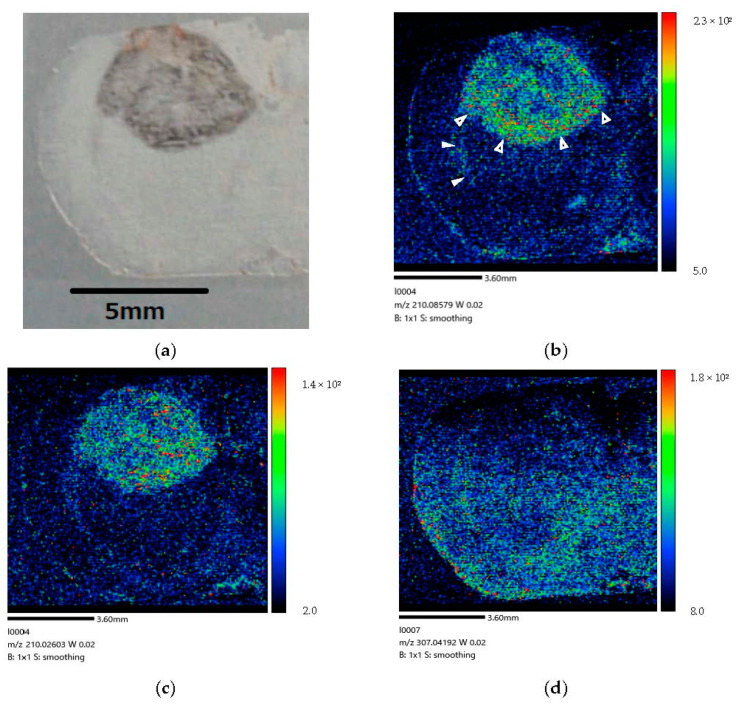
(**a**) Coronal section of brain-tumor-model rat after 160 min administration. (**b**) BPA biodistribution image generated by MALDI-MSI. BPA penetration is mainly visible in the brain tumor tissue. Some BPA is present in the lateral ventricle (➣). BPA is localized in the invasive margin (➤) of the brain tumor and clearly distinguishes the border between tumor and normal cells. These results imply that BPA penetrated brain tumor cells within 160 min following administration into the lateral ventricle of brain-tumor-model rat. (**c**) The mass image obtained from peaks at *m/z* 210.026 originated in tumor. The image showed a distribution different from the localization of BPA within the tumor. (**d**) The mass image obtained from peaks at *m/z* 307.042 originated from endogenous biomolecule.

**Figure 7 life-12-01786-f007:**
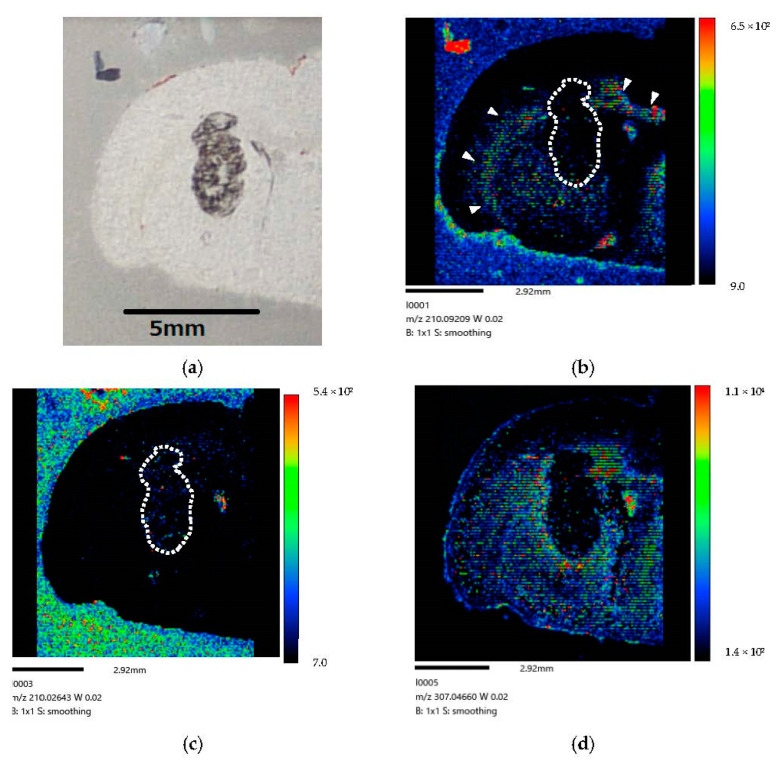
(**a**) Coronal section of rat-brain-tumor-model at 60 min post-injection. (**b**) BPA biodistribution image generated by MALDI-MSI. BPA is only visible along the lateral ventricle (➣) and not in tumors. The dotted line (…..) indicates where the tumor is expected to exist. A small amount of BPA is visible in normal tissue. Our results imply that BPA was excreted from brain tumor cells at 60 min post-infusion following administration into the lateral ventricle of brain-tumor-model rat. (**c**) The mass image obtained from peaks at *m/z* 210.026 originated in tumor. The dotted line (…..) indicates where the tumor is expected to exist. The image showed a distribution different from the localization of BPA within the tumor. (**d**) The mass image obtained from peaks at *m/z* 307.04 originated from endogenous biomolecule.

**Figure 8 life-12-01786-f008:**
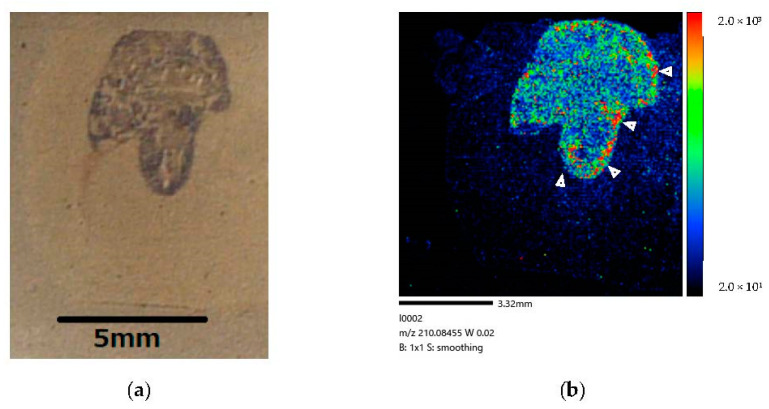
(**a**) Coronal section of brain-tumor-model rat after 120 min administration. (**b**) BPA biodistribution image generated by MALDI-MSI. BPA penetration is mainly observed in brain tumor tissue. No BPA is visible in the lateral ventricle. BPA is localized in the invasive margin (➤) of the brain tumor and clearly distinguishes the border between tumor and normal cells. Our results imply that BPA penetrated the brain tumor cells within 120 min following administration via blood circulation of the brain-tumor-model rat.

## Data Availability

Not applicable.

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
