# Peer review of "Boron Delivery to Brain Cells via Cerebrospinal Fluid (CSF) Circulation in BNCT of Brain-Tumor-Model Rats—Ex Vivo Imaging of BPA Using MALDI Mass Spectrometry Imaging"

_life, 2022, doi:10.3390/life12111786_

Round 1
Reviewer 1 Report
Thank you for submitting the manuscript. I think that this study would include significant results enough for publication. However, some aspects need to be addressed.
Materials and Methods
line 115-117 Is the introducing point of a guide cannula median or left-side or right-side?
In this study the authors used one animal for each condition (four conditions; CSF 30min, CSF 160min + immediately collect, CSF 160min + after 60min collect, IV administration). Are there any data to verify the reproducibility of these methods (each condition)?
Figures
Figure 1 The triangle marker is very obscure (I cannot see.).
Figure 1 and 3 There are no explanation for broken lines. The authors should describe what it indicates.
Discussion
Line 229-231 The authors estimate that over 60 min of drug administration is required for penetration throughout the entire brain tissue after infusion from these results of this study. What is the reason for the authors thinking that the main cause for the difference of BPA biodistribution is the time of drug administration, not the total volume of BPA, not the time after the start of the infusion.
How do the authors expect the result of the same experiment with the condition below?
condition 1; infusion at 20mg/kg/hr (30min)(total volume is nearly equal to 160min infusion), brain sample is collected at 0 min after the end of the infusion
condition 2; infusion at 4mg/kg/hr (30min), brain sample is collected at 30 min after the end of the infusion (at 60min from the start of infusion)
Author Response
Dear Reviewer1,
I appreciate for reviewing our paper. We made some modifications following your suggestion.
Your detailed comments gave a better indication of our research results.
Thank you very much for your valuable comments
Sachie Kusaka,
Osaka University, Osaka, Japan
Reviewer 2 Report
Sachie Kusaka et al. in the wake of their previous paper (Biology 2022, 11, 397) aimed to demonstrate the usefulness of the CSF delivery method for brain tumors, in the present work successfully investigated the biodistribution of BPA in rat brain after an infusion of BPA via the cerebrospinal fluid circulation. Obtaining evidence of BPA penetration into the brain parenchyma after administration via the cerebrospinal fluid. The work is very interesting with high applicative value, it is clearly written and the conclusions are consistent with the results presented.
I suggest the authors to address the following points which, in relation to their experience in the sector, can be considered minors but must be addressed before being accepted on Life MDPI
Introduction: it is clear, but it is short and is not sufficiently informative. It must be implemented in the first part also contemplating literature on: the opening techniques of the BBB (e.g., by ultrasound and UCA), the delivery of BPA coupled with carriers, and on the procedures of radioembolization via catheter, including and remarking the disadvantages related to biodistribution and toxicity if compared to CSF ​​delivery method.
Materials and methods: this session is sufficiently clear, but I would like there to be an additional paragraph dedicated to the statistical analysis of the results obtained: the number of rats used, the statistical solidity and the reproducibility of the data obtained must be clarified. Other biodistribution images generated by MALDI-MSI could also be added in a support material together with a sketch of the experimental setup. Moreover, even if it has already been published in the previous work, also for greater clarity the authors should add also in this work (also as supplementary material) an experimental scheme that summarizes the administration setup.
Results: Greater clarity for non-expert life readers could be provided by the addition of information on controls, concerning in particular BPA-free injections, Rats without tumor, and a comparison of biodistribution obtained by delivery via BBB using the same imaging methodology (even if only in reference to the literature).
Figures: the quality of panels (a) of all figures is not sufficient and scales are missing.
Discussion and conclusion: Discussion section 4.3 must be interlaced with the literature and expanded otherwise it must be incorporated into section 4.4. At the beginning of the discussion or in the conclusions, the work [7] must be mentioned, clarifying the degree of progress pointed out of the present work with respect to this work. The authors could support the discussion by recalling the scheme of the operational setup of delivery via CSF scheme (integrated and implemented I suppose) and confirming precisely on the basis of the encouraging results presented in this work.
Author Response
Dear Reviewer2,
I appreciate for reviewing our paper. I made some modifications following your suggestion.
Your advice gave us a lot of realizations and a good opportunity to think more deeply about our research. Thank you for your very valuable comments.
Sachie Kusaka,
Osaka University, Osaka, Japan

Reviewer 3 Report
The manuscript is well-written and the study concept is nice. The manuscript is acceptible in its current form.
Author Response
Dear Reviewer3,
I appreciate for reviewing our paper.
Sachie Kusaka,
Osaka University, Osaka, Japan
Round 2
Reviewer 2 Report
The authors answered satisfactorily to my requests. I fully understand the difficulty in producing statistically more robust data. This was now made clear in the manuscript. I believe that the results shown in the revised manuscript are valuable and the work is implemented to such an extent to deserve acceptance on Life.